# Targeting Esophageal Squamous Cell Carcinoma by Combining Copper Ionophore Disulfiram and JMJD3/UTX Inhibitor GSK J4

**DOI:** 10.3390/cancers15225347

**Published:** 2023-11-09

**Authors:** Canlin Yang, Fei Li, Yuanyuan Ren, Qianqian Zhang, Bo Jiao, Jianming Zhang, Junxing Huang

**Affiliations:** 1Department of Oncology, The Affiliated Taizhou People’s Hospital of Nanjing Medical University, Taizhou School of Clinical Medicine, Nanjing Medical University, Taizhou 225300, China; 15262732359@163.com (C.Y.); 15996005667@163.com (F.L.); 13912193344@139.com (Y.R.); 2Taizhou People’s Hospital Affiliated to Nanjing University of Chinese Medicine, Taizhou 225300, China; 3National Research Center for Translational Medicine (Shanghai), State Key Laboratory of Medical Genomics, Ruijin Hospital, Shanghai Jiao Tong University School of Medicine, Shanghai 200025, China; zhangqianmengjie@outlook.com (Q.Z.); bjiao@sibs.ac.cn (B.J.); 4Institute of Translational Medicine, Zhangjiang Institute for Advanced Study, Shanghai Jiao Tong University, Shanghai 201203, China

**Keywords:** disulfiram, JMJD3, UTX, UPR, ESCC

## Abstract

**Simple Summary:**

Targeted therapy for malignant esophageal squamous cell carcinoma (ESCC) remains a big challenge for our clinicians. In an effort to search for the vulnerability of ESCC, we applied a high-throughput drug-screening strategy and found that CuET, a copper chelation product of disulfiram, had a strong synergy effect with the JMJD3/UTX inhibitor GSK J4 in treating ESCC in vitro and in vivo. Interestingly, JMJD3/UTX’s diagnostic and prognostic value, as well as the underlying mechanisms associated with endoplasmic reticulum stress were revealed. Targeting JMJD3 and UTX in combination with disulfiram has the potential to provide a new safe, effective, and available therapy for ESCC.

**Abstract:**

The alcohol-averse drug disulfiram has been reported to have anti-tumor effects and is well suited for drug combinations. In order to identify potential drug combinations in esophageal squamous cell carcinoma (ESCC), we screened a bioactive compound library with the disulfiram copper chelation product CuET. The Jumonji domain-containing protein 3 (JMJD3) and the ubiquitously transcribed tetratricopeptide repeat protein X-linked (UTX) inhibitor GSK J4 were identified. To further understand the molecular mechanism underlying the efficient drug combination, we applied quantitative mass spectrometry to analyze the signaling pathway perturbation after drug treatment. The data revealed that the synergistic effect of GSK J4 and CuET was due to the interaction among JMJD3 and UTX, which may play important roles in maintaining endoplasmic reticulum (ER) homeostasis in tumor cells. Interestingly, our clinical data analysis showed that high expression of JMJD3 and UTX was associated with T stage and worse prognosis of ESCC patients, further supporting the importance of the above findings. In conclusion, our findings suggest that the combination of CuET and targeting JMJD3/UTX may be a safe, effective, and available treatment for ESCC.

## 1. Introduction

Esophageal carcinoma (EC) is the ninth most common cancer and the sixth most common cause of cancer-associated death globally [1,2,3]. Esophageal squamous cell carcinoma (ESCC) and esophageal adenocarcinoma (EAC) are the two major histological subtypes of EC. In Western countries, esophageal cancer is mainly adenocarcinoma, whereas in China, more than 90% of esophageal cancers are squamous cell carcinomas [1]. Despite continuous improvements in surgical techniques and chemoradiotherapy strategies, the 5-year survival rate of ESCC patients worldwide is still less than 30% [4]. In recent years, the programmed cell death protein 1 (PD-1) inhibitors camrelizumab and pembrolizumab have achieved promising results as second-line treatments for advanced ESCC [5,6], while most results with targeted therapeutics have not been confirmed. Therefore, it is necessary to explore novel biomarkers to better understand the tumorigenesis of EC and to develop new targeted therapeutic agents.

The growing demand for effective anti-tumor drugs has prompted researchers to search for potential drugs among those already approved by the Food and Drug Administration (FDA). Here, we were particularly interested in disulfiram (DSF). DSF, also known as the medication Antabuse, is an FDA-approved drug that has been used for more than 60 years as a treatment for alcohol dependence with well-established pharmacokinetics, safety, and tolerance at the US FDA-recommended dosage [7]. DSF was found to have a potent anti-cancer effect on several cancer types, such as lung cancer, breast cancer, glioblastoma, pancreatic cancer, and ESCC [8,9,10,11,12]. Inside the body, the DSF-reactive metabolite ditiocarb (DTC) forms complexes with copper—bis (diethyldithiocarbamate)-copper (also named CuET) [13,14]. It has been reported that CuET preferentially accumulates in tumor tissues and can bind to nuclear protein localization protein 4 (NPL4), an adaptor protein of the p97 segregase. The ubiquitin–proteasome system (UPS) is an important pathway for the degradation of most cellular proteins and plays a regulatory role in a variety of cellular processes. The p97 segregase plays a critical role in the UPS by separating poly-ubiquitinated substrates from their binding partners and presenting them to the proteasome for proteolysis [15]. CuET then inhibits the p97-dependent protein degradation pathway and triggers endoplasmic reticulum (ER) stress and the unfolded protein response (UPR) pathway, inducing cell apoptosis [14,15].

As an inexpensive and safe drug with relatively few side effects [13], DSF is well suited for addition to a combination regimen of chemoradiotherapy and targeted treatment. Combining drugs can reduce the dose and toxicity of each drug while maintaining the same drug efficacy, thus reducing toxicity to normal tissues. In the search for druggable targets of ESCC cell lines, we performed a cell-based cytotoxicity screen using an in-house compounds library as a chemical toolbox. We found that the combination of CuET and Jumonji domain-containing protein 3 (JMJD3) and ubiquitously transcribed tetratricopeptide repeat protein X-linked (UTX) inhibitor GSK J4 showed effective efficacy in the ESCC cell line TE10. JMJD3 and UTX are demethylases of histone H3 on lysine 27 (H3K27) that catalyze the expression of a series of genes by regulating the demethylation of H3K27m3 [16]. JMJD3 and UTX are involved in several biological processes in the human body and play important roles in physiological processes, such as cell differentiation, aging, embryonic development, and tumorigenesis. However, it remains highly controversial whether both play inhibitory or promotional roles in the tumorigenesis of different types of tumors [17,18]. The aim of our study is to show that combining CuET and JMJD3/UTX inhibitors may be a potential and promising therapeutic strategy for ESCC.

## 2. Materials and Methods

### 2.1. Compound Library and Drugs

The bioactive compound library, which contained 2149 compounds, including commercially available kinase inhibitors and a number of novel kinase inhibitors, was purchased from Selleck Company (Houston, TX, USA). These compounds are relatively potent and selective toward a relatively narrow array of kinase targets. CuET was a kind gift from the Shanghai Institute of Hematology. GSK J4 was purchased from BidePharm Company (Cat. #BD764815, Shanghai, China). 

### 2.2. Sampling

Patients with ESCC from eastern China who underwent surgical resection at Taizhou People’s Hospital from 2017 to 2020 and gave written informed consent before tissue collection were retrospectively reviewed. The retrospective study was approved by the Human Research Ethics Committee of Taizhou People’s Hospital. Patients with synchronous cancers of other organs and those who received preoperative therapy (including chemoradiotherapy, chemotherapy, radiotherapy, immune checkpoint inhibitor therapy, target therapy, and Chinese herbal therapy) were excluded. A total of 75 patients who underwent EC surgery were included in this study. Fresh paired tumor tissue and paraneoplastic tissue (at least 5 cm from the edge of tumor tissue) were collected. None of these patients received any anti-tumor treatment before surgery, and they were confirmed to have ESCC via postoperative pathology. The fresh specimens were transferred to liquid nitrogen immediately after collection. After 5 min, specimens were dispensed into RNA protective solution and placed in a −80 °C refrigerator for backup. The medical history and clinicopathological data of the 75 patients with ESCC were collected using the HIS system of Taizhou People’s Hospital. These patients and their families were regularly followed at 3-month intervals by telephone until 28 February 2022. The tumor–node–metastasis (TNM) staging of the patients was based on the American Joint Committee on Cancer (AJCC) 8th edition staging. Overall survival (OS) was calculated from the time of operation to death from any cause. Progression-free survival (PFS) was calculated from the time of operation to local recurrence and/or new distant metastases or death from any cause without evidence of recurrence and/or new distant metastases.

### 2.3. Cell Lines and Cell Culture

The ESCC cell lines TE10 (RRID: CVCL_1760) and KYSE410 (RRID: CVCL_1352) were purchased and characterized using short tandem repeat (STR) markers from Meixuan Company (Shanghai, China). All experiments were performed with mycoplasma-free cells. All the cell lines were cultured in an RPMI 1640 medium (Cat. #10-040-CRV, Corning, NY, USA) supplemented with 10% fetal bovine serum (FBS; Cat. #10099-141, Gibco, Grand Island, NY, USA), 100 U/mL penicillin, and 100 μg/mL streptomycin (Cat. #15140112, Gibco, Grand Island, NY, USA). Cells were incubated at 37 °C in a 5% CO_2_-95% air atmosphere.

### 2.4. Screening Bioactive Compound Library

We screened a bioactive compound library in the ESCC cell line TE10 to identify the potential efficient combination of drug therapy with CuET. Drug screening was performed at a concentration of 2 μM with or without CuET (0.2 μM) in a 48 h cellular proliferation assay with two technical replicates (Figure 1B). The combinatory drug effect was measured using CellTiter-Glo^®^ luminescent cell viability assay kit (Cat. #G7571, Promega, Madison, WI, USA) for automated high-throughput screening, cell proliferation, and cytotoxicity assays.

### 2.5. Drug Synergy Testing

Further validation of the drug synergism was performed in TE10 and KYSE410 cells. Cells were seeded into 96-well plates at 3000 cells/well, and six concentration gradients were set for each of the two drugs to form a matrix (*n* = 4). Cell viability was detected using the CellTiter-Glo^®^ luminescent cell viability assay kit (Cat. #G7571, Promega, Madison, WI, USA) after 48-h incubation in the incubator. The data were imported into the Calcusyn software to derive the combination index (CI), which was used to measure the combined effect of drugs. CI < 0.1, CI 0.1–0.3, CI 0.3–0.7, CI 0.7–0.85, CI 0.85–0.90, and CI 0.90–1.10 indicate very strong synergism, strong synergism, synergism, moderate synergism, slight synergism, and nearly additive effects, respectively. Furthermore, CI 1.10–1.20, CI 1.20–1.45, CI 1.45–3.3, CI 3.3–10, and CI > 10 indicate slight antagonism, moderate antagonism, antagonism, strong antagonism, and very strong antagonism, respectively [19]. Dose-reduction index (DRI) values were used to measure how many folds the dose of each drug in combination may be decreased at a given effect level when compared with the doses of each drug alone [20].

### 2.6. Annexin V Staining

ESCC cell apoptosis was assayed using the Annexin V method. After treatment, TE10 and KYSE410 cells were washed with precooled PBS, digested with EDTA-free trypsin, and then stained with an Annexin V-FITC/PI kit (Cat. #CA1020, Solarbio, Beijing, China) according to the manufacturer’s protocol. The cell apoptosis was analyzed using an EXFOLW-206 Flow Cytometry (DAKEWE, Shenzhen, China).

### 2.7. Cell Cycle Analysis

After drug treatment, TE10 and KYSE410 cells were harvested, washed using PBS three times, and then fixed in precooled 70% ethanol at 4 °C overnight. After incubation with propidium iodide (PI) solution (Cat. #C1052, Beyotime, Shanghai, China) for 30 min at 37 °C in the dark, the cells were measured using an EXFOLW-206 Flow Cytometry (DAKEWE, Shenzhen, China).

### 2.8. Quantitative Reverse Transcription Polymerase Chain Reaction (RT-qPCR)

Total RNA was extracted from specific tissues using an RNAeasy™ RNA isolation kit (Cat. #R0027, Beyotime, Shanghai, China). cDNA was synthesized using BeyoRT™ III First Strand cDNA Synthesis Master Mix (Cat. #D7185L, Beyotime, Shanghai, China). RT-qPCR was performed on a LightCycler 480 II (Roche, Basel, Switzerland) instrument using BeyoFast™ SYBR Green qPCR Mix (Cat. #D7260, Beyotime, China). The special primers were used as followed: JMJD3 (forward), 5′- CCCTGGAACGATACGAGTGG-3′; JMJD3 (reverse), 5′- TCTTGAACAAGTCGGGGTCG-3′; UTX (forward), 5′- TGGCCAATGGACCCTTTTCTG-3′; UTX (reverse), 5′- GGTCAGGTTTGTGCGGTTATG-3′; GAPDH (forward), 5′- GCACCGTCAAGGCTGAGAAC-3′; GAPDH (reverse), 5′- TGGTGAAGACGCCAGTGGA-3′. RT-qPCR conditions were 95 °C for 2 min and 45 cycles of 95 °C for 15 s, 60 °C for 20 s, and 72 °C for 30 s. All RT-qPCR experiments were performed in triplicates. RT-qPCR was performed to detect the mRNA expression levels of the target genes. ΔCt values were used to determine absolute expression, and ΔΔCt values were used to determine relative expression as fold changes occurred. Using the 2^−ΔΔCt^ method, the relative expression levels of the target genes for each sample were normalized to those of the endogenous control GAPDH.

### 2.9. Western Blot

Cells were lysed using radioimmunoprecipitation (RIPA) buffer (Cat. #P0013, Boyotime, Shanghai, China) supplemented with the protease inhibitor (Cat. #11697498001, Roche, Basel, Switzerland) and phosphatase inhibitors (Cat. #04906837001, Roche, Basel, Switzerland), sonicated, and incubated on ice for 30 min. Homogenates were centrifuged at 12,000 rpm for 5 min at 4 °C. Supernatants were collected, and protein concentrations were determined using BCA (Cat. #P0010, Boyotime, Shanghai, China). Equal amounts of 20 µg proteins were subjected to SDS-PAGE and then electrically transferred to polyvinylidene difluoride (PVDF) membranes. The membrane was blocked with 5% BSA for 1 h at room temperature and then incubated overnight at 4 °C with the primary antibodies. After washing three times with TBST (0.1% Tween in TBS), the membrane was incubated with secondary antibodies for 2 h at room temperature. The band intensity was estimated using Photoshop Software. The following antibodies were used in Western blot experiments: mouse anti-GAPDH (Cat. #AB8245, 1:5000, Abcam, Cambridge, MA, USA), rabbit anti-C-caspase-3 (Cat. #19677-1, 1:1000, Proteintech, Chicago, IL, USA), rabbit anti-eIF2α (Cat. #3398T, 1:5000, Cell Signaling Technology, Danvers, MA, USA), rabbit anti-IRE1 (Cat. #AB124945, 1:5000, Abcam, Cambridge, MA, USA), rabbit anti-ATF6 (Cat. #65880T, 1:5000, Cell Signaling Technology, Danvers, MA, USA), HRP-conjugated Goat Anti-Rabbit IgG (Cat. #D110058-0100, 1:5000, Sangon Biotech, Shanghai, China), HRP-conjugated Goat Anti-Mouse IgG (Cat. #D110087-0100, 1:5000, Sangon Biotech, Shanghai, China). GAPDH was used to probe each stripped membrane to verify the relative protein loading.

### 2.10. Quantitative Proteomics

Mass spectrometry experiments were performed at the Shanghai Key Laboratory of Regulatory Biology (School of Life Sciences, East China Normal University). Collected adherent cells were resuspended in a lysis buffer (8 M urea), followed by sonication on ice for 10 min and centrifugation at 16,000× *g* for 5 min at 4 °C. The supernatant protein concentration was quantified using BCA (Cat. #P0010, Boyotime, Shanghai, China), then 20 μg of protein was reduced with 10 mM of DTT at 55 °C for 30 min, alkylated with 15 mM iodoacetamide for 20 min, desalted with homemade desalination TIP, and resuspended in 20 µL of buffer A (2% acetonitrile, 0.1% formic acid in water) solution for mass spectrometry analysis. The MaxQuant (version 1.4.1.2) software was used for proteomic data analysis. Data were analyzed using R version 4.0.3 (R-Core-Team, 2020), and Gene Ontology (GO) analysis was performed using the package clussterProfiler. Proteins with a fold change greater than 1.5 or less than 0.67 and meeting a *p*-value less than 0.05 were defined as differential proteins. *p*-values were calculated using a *t*-test for differential proteins.

### 2.11. In Vivo Tumor Experiments

Female Balb/c nude mice were fed in the Taizhou people’s hospital animal laboratory with mimic normal diurnal. TE10 cells were injected (1 × 10^6^ cells were transplanted subcutaneously) to grow tumors in nude mice with a body weight of 18 ± 2 g and aged 4 weeks. After the tumors grew to 80~100 mm^3^ on average, 24 nude mice were randomly divided into three groups of eight mice each according to the different drug administration methods, as follows: (1) negative control group (normal diet and PBS intraperitoneal injection); (2) drug A group (normal diet plus oral 50 mg/kg DSF + 0.15 mg/kg copper gluconate (CuGlu) and PBS intraperitoneal injection); and (3) drug B group (normal diet plus oral 50 mg/kg DSF + 0.15 mg/kg CuGlu and 100 mg/kg GSK J4 intraperitoneally). The body weight and tumor size of nude mice were measured and recorded on the day of drug administration and periodically every three days. The tumor growth inhibition (TGI) was calculated using the formula TGI = (1 − *V*_Treated_/*V*_Control_), where *V*_Treated_ is the mean tumor volumes in the drug-treated group and *V*_Control_ is the mean tumor volumes in the control group. After the end of the administration, mice were sacrificed by cervical dislocation, and the tumors were extracted and frozen at −80 °C.

### 2.12. Histopathology and IHC Analyses

Tumor tissues dissected from mice were fixed in 4% Paraformaldehyde (PFA) for more than 24 h, followed by dehydration, clearing, and embedding in paraffin. Tissues were cut into 4 μm sections. The slides were de-paraffinized and rehydrated with gradient ethanol (100%, 95%, 85%, 75%, and 50%) and deionized water for 5 min, respectively. For antigen retrieval, the slides were placed in a container covered with sodium citrate buffer (10 mM and pH 6.0) and heated in a microwave for 15 min. Then, the slides were washed, blocked with 5% BSA for 30 min, and incubated with the appropriate dilutions of primary antibodies (C-caspase-3, Cat. #19677-1, 1:500, Proteintech, Chicago, IL, USA) overnight at 4 °C. After washing with PBS 5 times, the slides were incubated with Fluorescent secondary antibody (HRP-conjugated Goat Anti-Rabbit IgG, Cat. #D110058-0100, 1:100, Sangon Biotech, Shanghai, China) for 30 min at room temperature in the dark. The addition of DAPI was used to stain the nuclei for 2 min in the dark. Finally, the slices were sealed with glycerol and immediately observed under a fluorescence microscope. Positive cells were identified by the presence of brownish-yellow granules in the cytoplasm. Three high magnification (200×) fields of view were randomly selected for image acquisition in each section. Image Pro Plus 6.0 image analysis software was used for semi-quantitative analysis. The optical density (OD) values and percentage of positive areas of the selected three fields of view were detected. The magnitude of the average positive index (OD × percentage of positive area) was used to reflect the expression of C-caspase-3.

### 2.13. TUNEL Assay

Apoptotic cells were assessed via the terminal deoxynucleotidyl transferase-mediated dUTP nick-end labeling (TUNEL) technique using a TUNEL assay kit (Cat. #11684817910, Roche, Basel, Switzerland) following the manufacturer’s instructions. For the TUNEL staining of nuclei, the percentage of positive cells among at least 100 cancer cells from three randomly selected fields of vision observed using a high-power lens was calculated.

### 2.14. Statistical Analysis

All statistical analyses were performed using GraphPad Prism software version 6.01 (La Jolla, CA, USA). A Student’s *t*-test, analysis of variance, and chi-square test were used for statistical analysis. The CI value was calculated using CalcuSyn Version 2.0 (Biosoft, Ferguson, MO, USA) software. For survival analysis, survival curves were plotted using the Kaplan–Meier method and evaluated for statistical significance using a log-rank test. A *p*-value of less than 0.05 was considered statistically significant.

## 3. Results

### 3.1. Identification of Compounds Having Synergistic Effects with CuET

In order to search druggable targets of ESCC cell lines, we performed a cell-based cytotoxicity screen by using the combination of CuET and an in-house compounds library as a chemical toolbox. The CellTiter-Glo^®^ assay was used to assess the viability of the cells, and the results showed that CuET had an IC50 of 0.59 μM in the ESCC cell line TE10 (Figure 1A,B). After screening 2149 compounds from the Selleck bioactive compound library, we identified 167 molecule compounds that might have a synergistic effect with CuET in the ESCC cell line TE10 (Appendix A). The ratio of mean relative cell viability of single drug group and double drug group ≥ 1.2 and a difference ≥10% were set as the selection criteria. Among the top hits, the JMJD3/UTX inhibitor GSK J4 had a strong synergistic effect with CuET.

### 3.2. JMJD3 and UTX Were Highly Expressed in ESCC

We evaluated the mRNA expression of JMJD3 and UTX in primary ESCC specimens from 75 patients from eastern China, including 55 (73.3%) men and 20 (26.7%) women, whose mean age was 66 (range 50–79) years. The 8th AJCC stages of 75 patients were stage I–II in 41 (54.67%) patients and stage III–IV in 34 (45.3%) patients. Other clinicopathologic characteristics are provided in Table 1. The mean follow-up time was 28.1 (range 1–56) months. In addition, 34 (45.3%) patients developed local recurrence and/or new distant metastases during the follow-up. Specific clinicopathological data (including perioperative treatments, surgical operations, follow-up, etc.) are presented in Appendix A. 

The mRNA level of JMJD3 was significantly highly expressed in ESCC tissues (Z = −4.33, *p* < 0.01) using Wilcoxon signed rank sum test analysis in comparison to paraneoplastic tissues (Figure 2A). The mRNA level of UTX also showed significantly higher expression in ESCC tissues compared with paraneoplastic tissues (Z = −3.332, *p* < 0.01) (Figure 2B). In addition, there was a significant positive correlation between the relative mRNA expression of JMJD3 and UTX in ESCC (Spearman correlation coefficient = 0.831, *p* < 0.01) (Figure 2C).

We used the median mRNA expression levels of JMJD3 and UTX as two cut-off points and then divided 75 ESCC patients into a JMJD3 high expression group (*n* = 38), JMJD3 low expression group (*n* = 37), UTX high expression group (*n* = 37), and UTX low expression group (*n* = 38). As shown in Table 1, the clinicopathological factors of the patients, such as gender, age, tumor length, tumor location, degree of tumor differentiation, T-stage of the tumor, presence or absence of lymph node metastasis (LNM), TNM stage, and mRNA expression levels of JMJD3 and UTX were subjected to chi-square test. The results showed that the mRNA expression levels of JMJD3 and UTX were significantly correlated with the T-stage of ESCC, i.e., the depth of tumor infiltration (JMJD3: χ^2^ = 6.477, *p* = 0.011; UTX: χ^2^ = 5.856, *p* = 0.016), whereas no significant correlation was found with the other clinicopathological factors.

### 3.3. Expression of JMJD3 and UTX Is Associated with the Prognosis of ESCC Patients

The association of JMJD3 and UTX expression with OS and PFS in ESCC patients was analyzed using Kaplan–Meier survival curves and log-rank tests, respectively. As shown in Figure 2, high mRNA expression of JMJD3 was significantly associated with worse OS (median OS: low JMJD3 expression group not reached VS high JMJD3 expression group 30 months, HR = 0.44, 95%CI: 0.21–0.89, *p* = 0.025) (Figure 2D) and shorter PFS (median PFS: low JMJD3 expression group not reached VS high JMJD3 expression group 18 months, HR = 0.47, 95%CI: 0.23–0.90, *p* = 0.027) (Figure 2E) in ESCC patients, respectively. Similarly, high mRNA expression of UTX was significantly associated with worse OS (median OS: low UTX expression group not reached VS high UTX expression group 27 months, HR = 0.47, 95%CI: 0.22–0.94, *p* = 0.036) (Figure 2F) and shorter PFS (median PFS: low UTX expression group not reached VS high UTX expression group18.0 months, HR = 0.42, 95%CI: 0.20–0.79, *p* = 0.01) (Figure 2G) in ESCC patients, respectively.

### 3.4. The Combination Effect of CuET with JMJD3/UTX Inhibitor GSK J4

The strong correlation between JMJD3/UTX expression and clinical prognosis of ESCC patients prompted us to further investigate the effect of the drug combination on ESCC tumor cells in vitro, which were represented by TE10 and KYSE410, respectively. As shown in Table 2, the CI value of CuET and GSK J4 combination in TE10 cells was 0.21 at IC50 level and even less than 0.1 at IC75 and IC90 levels, presenting a strong or very strong synergism. The CI value in KYSE410 cells was 0.67 at the IC50 level, 0.33 at the IC75 level, and 0.17 at the IC90 level, presenting a synergism or strong synergism. In addition, the doses of each drug were significantly reduced by 2-fold to one hundredfold when compared with that of single regiment treatment. As shown in Figure 1D,E, the cytotoxicity of the combination of 0.4 μM CuET/4 μM GSK J4 (*F* = 25.08, both *p* < 0.01) and 0.8 μM CuET/8 μM GSK J4 (*F* = 130.6, both *p* < 0.01) was significantly stronger than that of the single drug in TE10 cells, and in KYSE410 cells, the cytotoxicity of the combination of 0.2 μM CuET/2 μM GSK J4 (*F* = 42.79, both *p* < 0.01) was significantly more cytotoxic than the single drug. As shown in Figure 1F,G, we used the Bliss model to evaluate the synergistic effect of the drugs [21]. The blue area represents the synergistic effects of the two drugs.

### 3.5. CuET Combined with GSK J4 Promotes Apoptosis and Cell Cycle Arrest in ESCC Cells

Due to the strong cytotoxicity of the drug combination and the significant increase in the proportion of apoptotic/necrotic cells with the prolonged drug treatment, highly toxic concentrations of the drugs were chosen to incubate for 12 h to avoid massive cell death that would significantly affect experimental data collection on cell cycle and signaling pathways. The results of the Annexin V/PI assay showed that compared to the control (DMSO) group, TE10 (Figure 3A), and KYSE410 (Figure 3B) cells exhibited significant apoptosis (TE10, *F* = 166.35, both *p* < 0.01; KYSE410, *F* = 151.97, *P*_CuET_ = 0.012, *P*_GSK J4_ < 0.01) after treatment with 1 μM CuET or 20 μM GSK J4 for 12 h, respectively. When two drugs were combined, the apoptosis of TE10 and KYSE410 cells was significantly enhanced compared to that in the single drug group (TE10, *F* = 166.35, both *p* < 0.01; KYSE410, *F* = 151.97, both *p* < 0.01).

Both TE10 (Figure 3C) and KYSE410 (Figure 3D) cells showed a significant increase in the G2 phase (TE10, *F* = 142.05, *p* < 0.01; KYSE410, *F* = 60.77, *p* < 0.01) and a significant decrease in the G1 phase (TE10, *F* = 180.87, *p* < 0.01; KYSE410, *F* = 161.54, *p* < 0.01) after treatment with 1μM CuET for 12 h, suggesting that ESCC cells were blocked in the G2 phase by the effect of CuET. In contrast, TE10 and KYSE410 cells were significantly increased in the G1 phase (TE10, *F* = 180.87, *p* = 0.025; KYSE410, *F* = 161.54, *p* < 0.01) and decreased in the S phase (TE10, *F* = 48.35, *p* < 0.01; KYSE410, *F* = 212.83, *p* < 0.01) after treatment with 20 μM GSK J4 for 12 h, suggesting that ESCC cells were blocked in the G1 phase by the effect of GSK J4. These results indicated that both CuET and GSK J4 could induce cell cycle arrest in ESCC cells and have different impacts on the cell cycle, suggesting a potential mechanistic explanation for the synergistic effect of the drug combination.

As shown in Figure 3F, compared to the negative control group (DMSO), TE10 cells showed no significant change in C-caspase-3 expression after treatment with 1 μM CuET for 12 h, whereas C-caspase-3 expression increased significantly after treatment with 20 μM GSK J4 alone or drug combination (*F* = 21.88, *P*_GSK J4_ = 0.011, *P*_combination_ < 0.01). However, no significant change in C-caspase-3 expression was observed in KYSE410 cells either with single-agent treatment with CuET and GSK J4 or drug combination compared with the negative control (Figure 3G).

### 3.6. Proteomic Analysis of ESCC Cells Treated with CuET and GSK J4

Next, we extended our study to the mechanistic understanding of molecular events after the combined treatment of TE10 cells via proteomic analysis. TE10 cells were divided into four groups according to different drug treatments for non-labeled quantitative proteomics analysis. The specific dosing methods were as follows: group A, 1 μM CuET; group B, 20 μM GSK J4; group A + B, 1 μM CuET + 20 μM GSK J4; and group C (Control), DMSO. The drug treatment time was 12 h for all the groups. Three biological replicates were set up for each group. A total of 3130 proteins were quantified in the overall proteomic data, among which 2228 proteins were quantified in all three biological repeats (Appendix A). Ninety percent of the peptides had mass errors within ±1.25 PPM, suggesting a high confidence in the quantitative results (Appendix A). The correlation coefficients between all the groups of samples were higher than 0.90, indicating good reproducibility (Appendix A). By plotting box plots of the corrected protein signal intensities, it was found that the signal distribution was consistent among the samples, indicating better accuracy of protein quantification (Appendix A). Principal component analysis (PCA) revealed good separation among the four groups (Appendix A), indicating that the difference between the protein groups was obvious. Three biological replicates of each group were clustered in one place, suggesting a good reproducibility of this experiment. The number of differential proteins between groups was presented as a volcano plot (Figure 4A–D).

According to GO analysis, we found that the proteins upregulated in TE10 cells after CuET addition were mainly involved in ER stress, UPR, and autophagy-related pathways (Figure 4E). The upregulated proteins after the addition of GSK J4 were mainly involved in the pathways related to cellular energy metabolic processes (Figure 4F). The upregulated proteins after the combination of CuET and GSK J4 were also mainly involved in the pathways related to ER stress and UPR (Figure 4G). The proteins down-regulated in TE10 cells after the addition of CuET were mainly involved in pathways related to cellular energy metabolism (Appendix A), while the down-regulated proteins after the addition of GSK J4 were mainly involved in pathways related to ribosome synthesis and RNA metabolism (Appendix A). The down-regulated proteins after the combination of CuET and GSK J4 could not be enriched to yield plausible signaling pathways.

Compared with the CuET group, the proteins upregulated in TE10 cells after the treatment with CuET and GSK J4 were mainly involved in pathways related to cellular energy metabolism (Appendix A), whereas the down-regulated proteins were mainly involved in pathways related to ER stress, UPR, and heat shock response (HSR) (Figure 4H). Based on the heat map of factors related to the UPR pathway (Figure 4I), we found the elevated expression of many heat shock protein family members, such as HSPA1A, HSPA6, HSPA8, HSPB1, DNAJB1, and the rest of the proteins involved in protein folding function-related proteins in TE10 cells after the addition of CuET alone. Although no significant effects of GSK J4 alone on ER stress or UPR-related pathways were observed in the GO analysis results, it was still evident from the heat map that GSK J4 alone could also induce a series of elevated expression of proteins involved in ER stress and protein folding pathways, such as TOR1B, ERO1A, HSPA5, and HSPA9, which were somewhat complementary to the UPR-related proteins induced by CuET. Surprisingly, compared to the single-agent CuET group, the expression of many UPR-related proteins significantly decreased after the treatment of CuET combined with GSK J4. These data suggest that the activation of UPR induced by CuET was not enhanced after the combination of CuET and GSK J4 but showed a certain degree of inhibition.

### 3.7. CuET in Combination with GSK J4 Inhibits the Activation of UPR Pathway

Western blotting was used to detect trends in the expression of key proteins in the UPR pathway after drug action. The results showed that the expression of ATF6 was decreased in TE10 cells after the addition of 1 μM CuET or 20 μM GSK J4 alone compared to the negative control (DMSO), but the difference was not statistically significant (*F* = 16.44, *p* > 0.05). In contrast, after the addition of CuET combined with GSK J4, the expression of ATF6 was significantly decreased compared to that in the negative control and the single drug groups (*F* = 16.44, all *p* < 0.05) (Figure 4J,K). TE10 cells showed no significant change in p-eIF2α expression after the addition of CuET or GSK J4 alone compared with the negative control, whereas after the simultaneous addition of CuET and GSK J4, the expression of p-eIF2α was significantly decreased compared with the negative control and the single drug group (*F* = 12.22, all *p* < 0.01) (Figure 4J,L). The expression of p-IRE1 was significantly higher after the addition of CuET alone than in the negative control (*F* = 101.1, *p* < 0.01), whereas no significant change in p-IRE1 expression was observed after the addition of GSK J4 alone. In contrast to the enhanced expression effect of CuET, the expression of p-IRE1 significantly decreased after the simultaneous addition of CuET and GSK J4 compared with CuET alone (*F* = 101.1, *p* < 0.01) (Figure 4J,M).

These results indicate that CuET enhanced the expression of p-IRE1 in TE10 cells, thus activating the UPR signaling pathway and enhancing the ER protein folding ability of tumor cells. In contrast, GSK J4 alone did not affect the key proteins of the UPR signaling pathway. When CuET combined with GSK J4 acted simultaneously, the expression of ATF6 and p-eIF2α was significantly decreased, and the enhanced expression of p-IRE1 by CuET was also significantly inhibited, suggesting that CuET combined with GSK J4 could inhibit the activation of the UPR signaling pathway in TE10 cells.

### 3.8. DSF/CuGlu in Combination with GSK J4 Inhibits ESCC Growth In Vivo

We chose a xenograft mouse model to validate the synergistic effects of DSF/CuGlu and GSK J4 in vivo. We chose to feed disulfiram and copper gluconate to mice as the metabolism of disulfiram in vivo to chelate copper ions to form CuET, and the doses of disulfiram and copper gluconate were based on Skrott et al. [14]. As shown in Figure 5C, the mice in group B (DSF/CuGlu + GSK J4) had significantly higher body weights than those in the control group (Time: *F* = 770.6, *p* < 0.01; Time*Group: *F* = 10.81, *p* < 0.01; Between-Subjects effect: *F* = 3.95, *p* = 0.035; *p* = 0.03). However, the mice in group A (DSF/CuGlu) did not have a significant weight advantage over the control group. On day 28 after administration, the tumor growth inhibition rate was 35% in group A (DSF/CuGlu) and 55% in group B (DSF/CuGlu + GSK J4) compared to the negative control group (Time: *F* = 360.2, *p* < 0.01; Time*Group: *F* = 20.77, *p* < 0.01; Between-Subjects effect: *F* = 29.90, *p* < 0.01; both *p* < 0.01) (Figure 5D). The tumor growth inhibition rate of DSF/CuGlu combined with GSK J4 was significantly higher than that of DSF/CuGlu (*p* = 0.033). As shown in Figure 5E, the tumor weights in group A (DSF/CuGlu) and group B (DSF/CuGlu + GSK J4) were significantly lower than those in the negative control group (*F* = 64.09, both *p* < 0.01). The tumor weight in group B (DSF/CuGlu + GSK J4) was significantly lower than that in group A (DSF/CuGlu) (*F* = 64.09, *p* = 0.037). These data suggested that either DSF/CuGlu alone or DSF/CuGlu combined with GSK J4 could effectively inhibit the growth of ESCC, and the tumor suppression effect of the drug combination was significantly better than that of DSF/CuGlu alone.

The results of the TUNEL assay showed that both group A (DSF/CuGlu) and group B (DSF/CuGlu + GSK J4) exhibited a significant increase in the proportion of apoptotic tumor cells (*F* = 169.2, both *p* < 0.01), and DSF/CuGlu combined with GSK J4 promoted apoptosis significantly better than DSF/CuGlu alone (*F* = 169.2, *p* < 0.05) (Figure 5F,H).

The results of immunohistochemical experiments showed that compared with the negative control group, both group A (DSF/CuGlu) and group B (DSF/CuGlu + GSK J4) exhibited significantly higher expression of C-caspase-3 (*F* = 1577.0, both *p* < 0.01), and the elevated expression of C-caspase-3 after DSF/CuGlu + GSK J4 treatment was significantly better than that of DSF/CuGlu alone (*F* = 1577.0, *p* < 0.01) (Figure 5G,I). These data suggest that either DSF/CuGlu alone or a combination of DSF/CuGlu and GSK J4 can induce apoptosis in ESCC, and the apoptosis induced by the drug combination was stronger than that induced by DSF/CuGlu alone.

## 4. Discussion

It has been reported that oral administration of disulfiram and copper ions could result in preferential accumulation of CuET in tumor tissues and induce upregulation of the UPR-related proteins XBP1s, ATF4, and p-eIF2α, as well as a robust heat shock response in a variety of tumors [14]. The UPR pathway consists of three ER transmembrane proteins, PERK, IREl, and ATF6, and eIF2α is a downstream factor of PERK [22]. Under ER stress conditions, BiP (also known as HSPA5 and GRP78) segregates from the three sensors, PERK, IREl, and ATF6, and binds to misfolded proteins accumulated in the ER lumen, thereby activating the three ER stress sensors [22]. ER stress signaling activates the UPR signaling pathway, which helps restore the protein processing capacity and redox homeostasis of the ER, allowing cells to adapt to the ER stress environment, survive the stress period, and maintain protein homeostasis, thereby prolonging cell survival [22,23,24]. Heat shock response is another important way to maintain cellular protein homeostasis. Heat shock proteins, also known as molecular chaperones, can aid in the proper folding of proteins and inhibit the production of irreversible polymers [25]. However, when misfolded proteins continuously accumulate in the ER to a certain extent, the ER will be irreversibly damaged and its function cannot be restored, thus initiating apoptotic signaling pathways [23]. Indeed, our data showed that DSF/Cu could induce an autophagy-dependent apoptotic pathway in ESCC cells by activating the IRE1 pathway, thereby activating the UPR pathway.

JMJD3 can affect tumor cell stemness, proliferation, metastasis, apoptosis, and sensitivity to therapy by enhancing the transcriptional activity of C-MYC and activating the RAS/MEK pathway [26]. The high expression of JMJD3 and C-MYC is associated with poor prognosis in ESCC patients [17]. However, in other tumor types, JMJD3 has been shown to exert tumor-suppressive effects. JMJD3 knockdown enhances the proliferation of colorectal cancer cells by promoting cell cycle progression and inhibiting apoptosis, and low JMJD3 expression is an independent predictor of poor prognosis in patients after colorectal cancer surgery [27]. JMJD3 is expressed at significantly low levels in breast cancer tissues and is associated with poor prognosis in patients with breast cancer. Overexpression of JMJD3 can inhibit proliferation, infiltration, migration, and epithelial–mesenchymal transition (EMT) of breast cancer cells by decreasing the expression of β-catenin [28]. UTX has also been shown to play a bidirectional regulatory role in different tumor types. A study on breast cancer showed that UTX knockdown significantly reduced the proliferation and invasion of breast cancer cells in vitro and in vivo, and high UTX expression was associated with a worse prognosis in breast cancer patients [29]. Another study that collected 224 breast cancer patients showed that high UTX expression was significantly associated with high histological grade, lymph node invasion, vascular infiltration, and MMP-11 expression in breast cancer. Patients with high UTX expression had lower overall survival and progression-free survival [30]. An early study showed that re-transfection of the UTX wild-type gene into UTX gene-deficient ESCC cells resulted in reduced growth of ESCC cells, suggesting that UTX is a tumor suppressor [31]. These findings indicate that the pathophysiological functions of JMJD3 and UTX are highly context-dependent. In our collection of 75 ESCC tissue specimens, JMJD3 and UTX mRNA levels were significantly elevated, and the expression levels of JMJD3 and UTX were significantly associated with the depth of ESCC infiltration, worse OS, and shorter PFS in ESCC patients. These data suggest that JMJD3 and UTX may be potential diagnostic and prognostic molecular markers of ESCC.

GSK J4 is a potent and specific inhibitor of JMJD3 and UTX [32]. Recent studies have shown that GSK J4 affects the proliferation and apoptosis of various cancer cells. In high-risk neuroblastoma, GSK J4, in combination with vincristine, can effectively induce the differentiation and endoplasmic reticulum stress of neuroblastoma cells [33]. A recent study showed that GSK J4 induced apoptosis of human acute myeloid leukemia cell line KG-1a and upregulated the expression of ER stress-related proteins caspase-12, GRP78, and ATF4, while the apoptosis and cycle blocking induced by GSK J4 were significantly inhibited after the addition of the ER stress inhibitor 4-PBA [34]. These findings suggest that under ER stress, GSK J4 can induce the ER-associated apoptosis signaling pathway, whereas, without ER stress, the ability of GSK J4 to induce tumor apoptosis is obviously reduced. Therefore, GSK J4-induced apoptosis is ER stress-dependent. In our study, the GO analysis of mass spectrometric detection of differential genes for GSK J4 action was not enriched for ER stress or UPR-related pathways, and the Western blot results showed that GSK J4 did not cause significant changes in the expression of ATF6, p-eIF2α, and p-IRE1.

GSK J4 alone was less efficient in killing ESCC cells in vitro (Figure 1D,E). However, when CuET induced ER stress, the cytotoxicity of GSK J4 increased ten to hundredfold (Table 2). This phenomenon that the effectiveness of GSK J4 depends on ER stress suggests that H3K27m3 induced by GSK J4 can silence ER-relevant genes’ expression, thus leading to sustained cellular damage. The GO analysis and heat map of the UPR pathway showed that the activation of UPR induced by CuET was not enhanced after the drug combination but showed a certain degree of inhibition, especially a large number of heat shock family proteins (Figure 4). In addition, Western blot results showed that the expression of both ATF6 and p-eIF2α was significantly decreased, and the enhanced expression of p-IRE1 by CuET was also significantly inhibited when ESCC cells were treated by combining CuET and GSK J4. These data suggest that after the activation of ER stress and UPR induced by CuET, the addition of GSK J4 can inhibit the activation of the UPR pathway. Therefore, the ER cannot enhance the ability of cells to process misfolded proteins, resulting in the continuous accumulation of misfolded proteins in the ER, causing irreversible damage to the ER, thus activating the ER-associated apoptotic pathway, which may be one of the reasonable mechanisms to account for the potent synergistic effect between CuET and GSK J4 (Figure 6). JMJD3 and UTX may play important roles in maintaining ER homeostasis in tumor cells. However, whether blockage of JMJD3/UTX-induced H3K27m3 directly downregulated the expression of UPR-related genes or affected other upstream genes remains unclear, and further experiments are needed to clarify this in the future.

## 5. Conclusions

In summary, CuET has a strong synergistic effect with JMJD3/UTX inhibitor GSK J4 in treating ESCC both in vitro and in vivo. JMJD3 and UTX may play important roles in maintaining ER homeostasis in tumor cells. Furthermore, patients with high JMJD3 and UTX expression showed more pronounced tumor infiltration and possessed worse OS and PFS. Targeting JMJD3 and UTX in combination with disulfiram has the potential to provide a new, safe, effective, and available therapy for ESCC.

## Figures and Tables

**Figure 1 cancers-15-05347-f001:**
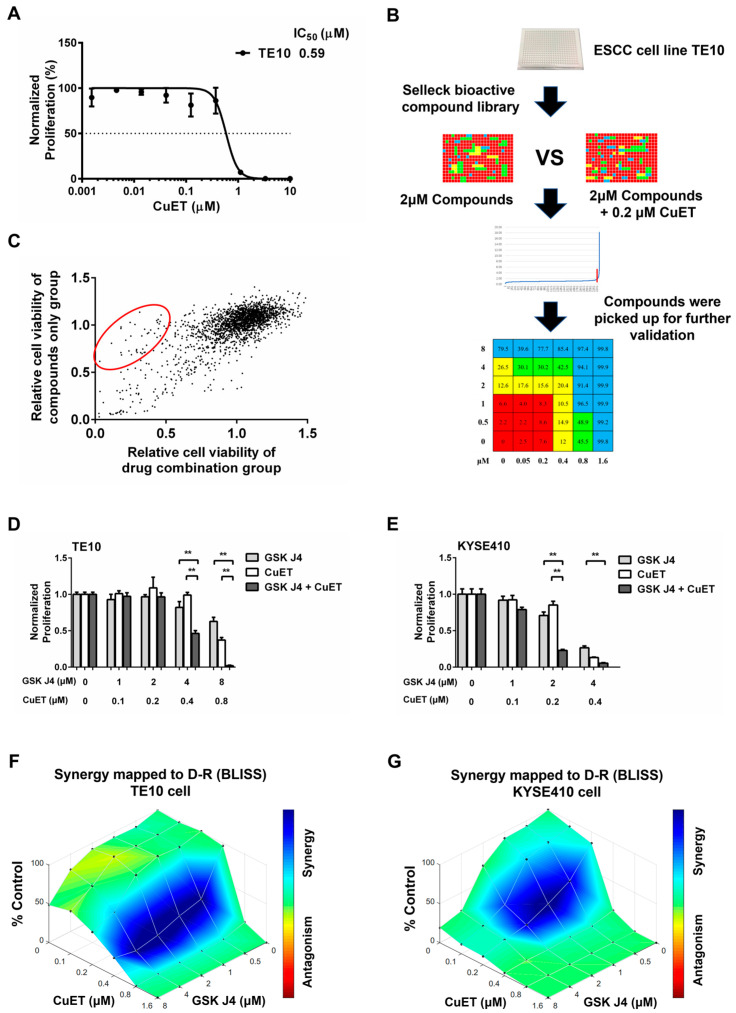
The combination of CuET and GSK J4 had a strong synergistic effect. (**A**) The IC50 value of compound CuET in the ESCC cell line TE10 was 0.59 μM (*n* = 6). Cell viability was assayed using the CellTiter-Glo^®^ assay. (**B**) Flow chart of high-throughput combination drug screening. (**C**) Relative cell viability of compounds in the single or drug combination groups. The red region indicates compounds showed synergistic effects with CuET. (**D**) Cell viability of TE10 after the addition of different concentrations of CuET and GSK J4 for 48 h (*n* = 4). (**E**) Cell viability of KYSE410 after the addition of different concentrations of CuET and GSK J4 for 48 h (*n* = 4). (**F**,**G**) Bliss dose-effect surface model of CuET and GSK J4 in TE10 (**F**) and KYSE410 (**G**) cells. Blue areas indicate synergistic effects of CuET and GSK J4. **, *p* < 0.01.

**Figure 2 cancers-15-05347-f002:**
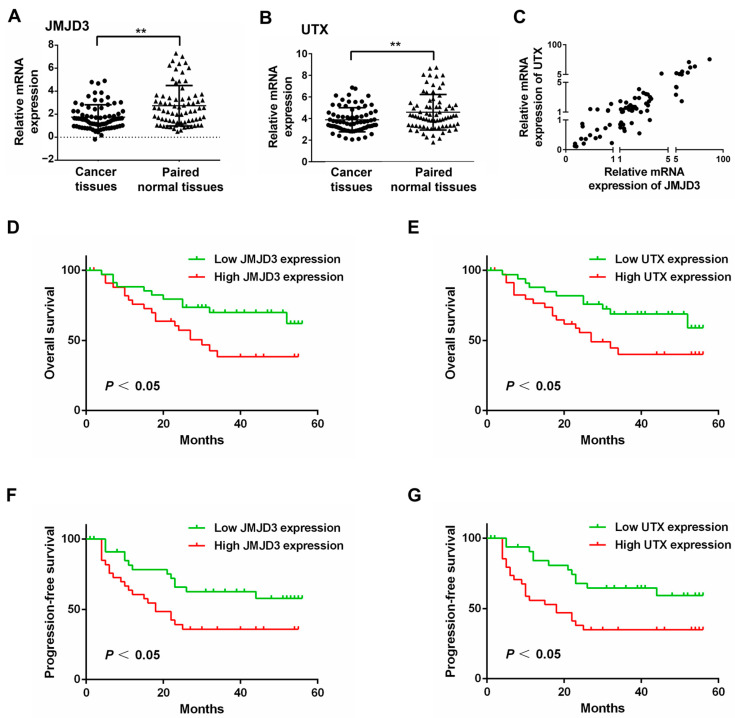
High expression of JMJD3 and UTX was associated with poor survival of ESCC. (**A**) mRNA expression of JMJD3 in ESCC tissues and paraneoplastic tissues (Wilcoxon signed rank sum test; Z = −4.33; **, *p* < 0.01). (**B**) mRNA expression of UTX in ESCC tissues and paraneoplastic tissues (Wilcoxon signed rank sum test; Z = −3.332; **, *p* < 0.01). (**C**) The correlation of the relative mRNA expression of JMJD3 and UTX (Spearman correlation coefficient = 0.831; **, *p* < 0.01). (**D**,**E**) Kaplan–Meier survival curves for overall survival of ESCC patients with different JMJD3 expression (**D**) and UTX expression (**E**). (**F**,**G**) Kaplan–Meier survival curves for progression-free survival of ESCC patients with different JMJD3 expression (**F**) and UTX expression (**G**).

**Figure 3 cancers-15-05347-f003:**
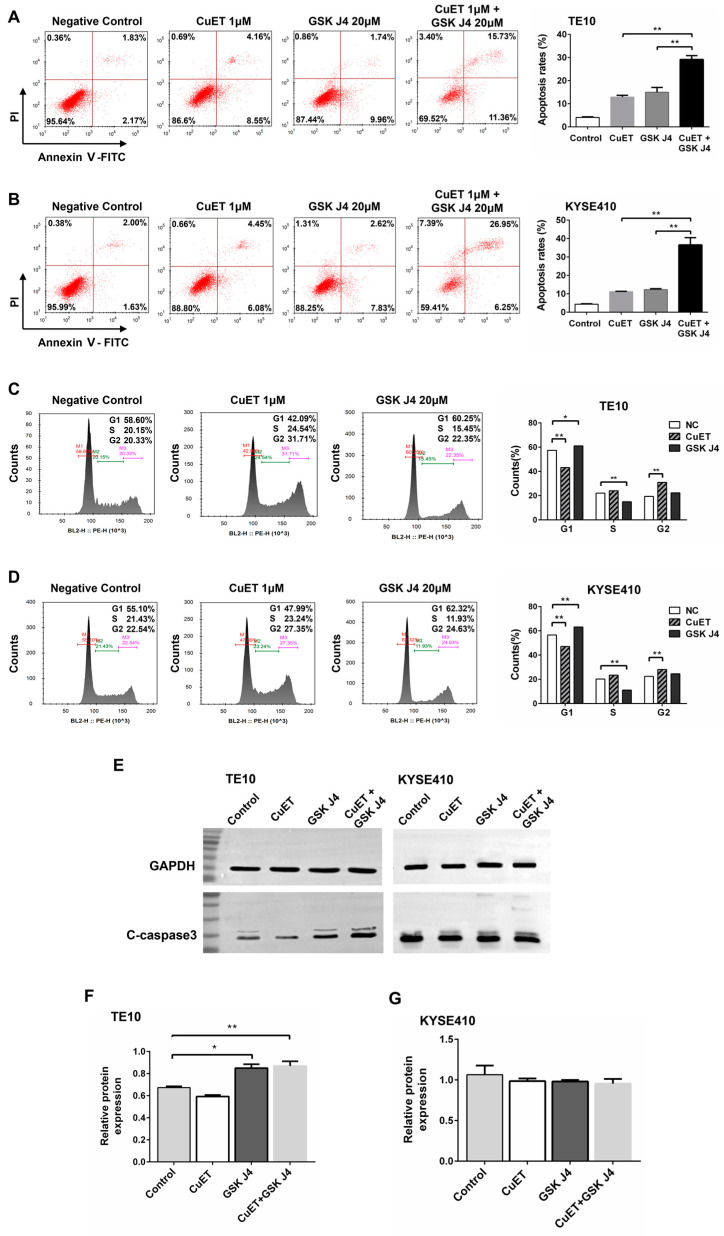
The combination of CuET and GSK J4 promoted apoptosis and cycle arrest in ESCC cells. (**A**,**B**) Annexin V/PI assay was performed to detect the apoptotic effect of CuET and GSK J4 on TE10 (**A**) and KYSE410 cells (**B**) (*n* = 3). (**C**,**D**) Effects of CuET and GSK J4 on the cell cycle of TE10 (**C**) and KYSE410 cells (**D**) (*n* = 3). The combination of CuET and GSK J4 significantly enhanced C-caspase-3 expression in TE10 cells. Original images can be found in Appendix A (**E**) Western blot assay to detect the effect of CuET and GSK J4 on the protein expression of C-caspase-3 in TE10 and KYSE410 cells (*n* = 3). (**F**,**G**) Statistical analysis of the Western blot results in (**E**). C-caspase-3, cleaved caspase-3; *, *p* < 0.05; **, *p* < 0.01.

**Figure 4 cancers-15-05347-f004:**
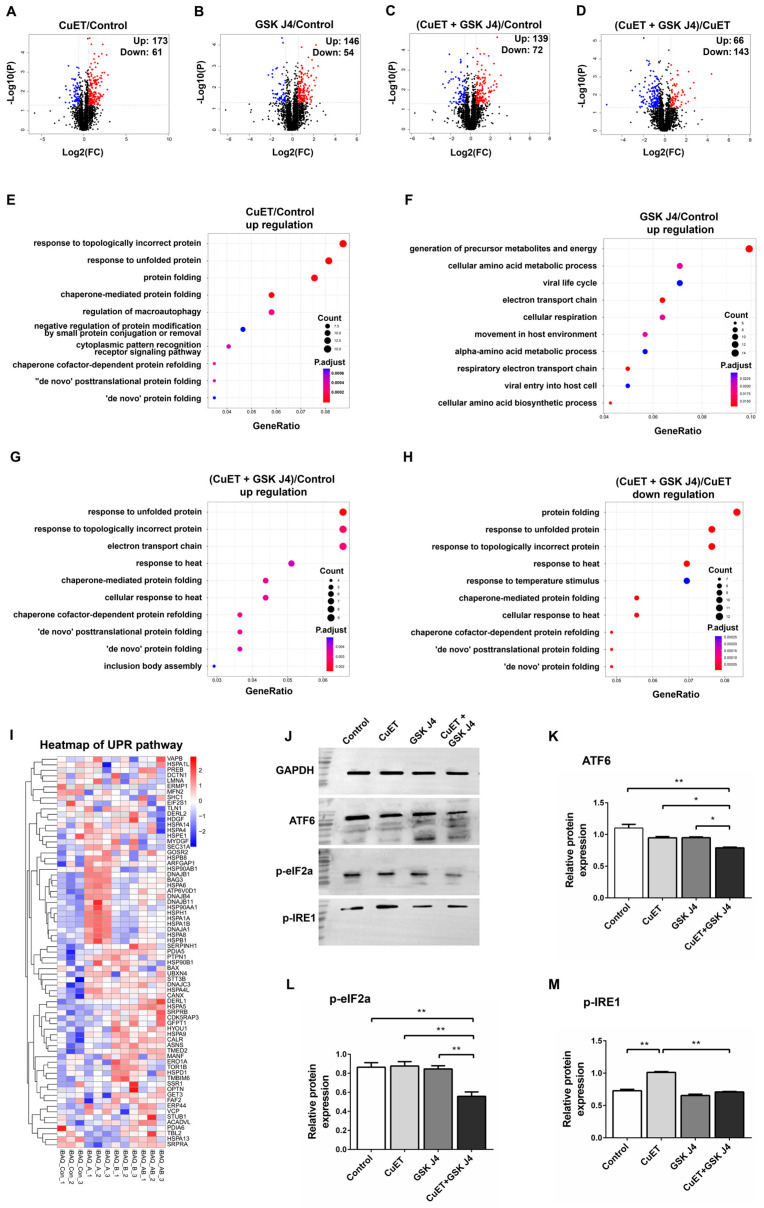
Proteomic analysis of TE10 cells treated with CuET and GSK J4. (**A**–**D**) Volcano plot of differentially expressed proteins. (**A**): the negative control (DMSO) VS group A (CuET); (**B**): the negative control VS group B (GSK J4); (**C**): the negative control VS group A + B (CuET + GSK J4); (**D**): group A (CuET) VS group A + B (CuET + GSK J4). Red dots in the graph indicate upregulated proteins, and blue dots indicate down-regulated proteins. (**E**–**G**) GO enrichment bubble plot of upregulated proteins between different groups. (**E**): group A (CuET) compared to the control group. (**F**): group B (GSK J4) compared to the control group. (**G**): group A + B (CuET + GSK J4) compared to the control group. (**H**) GO enrichment bubble plot of down-regulated proteins between group A + B (CuET + GSK J4) compared to group A (CuET). A larger bubble area indicates that more genes are involved in the signaling pathway, and a redder bubble color indicates a smaller *p*-value and higher confidence. (**I**) Heat map of the distribution of proteins associated with the UPR pathway. A: group A (CuET); B: group B (GSK J4); Con: group C (DMSO); AB: group A + B (CuET + GSK J4). The color intensity of the squares indicated the protein expression level, red and blue for high and low expression, respectively. (**J**) Western blot assays detected the effect of different treatments (DMSO, 1 μM CuET, 20 μM GSK J4, and 1 μM CuET + 20 μM GSK J4) for 12 h on the expression of ATF6, p-eIF2α, and p-IRE1 in TE10 cells (*n* = 3). Original images can be found in Appendix A. (**K**–**M**) Statistical analysis of the Western blot results in (**J**). (**K**): ATF6; (**L**): p-eIF2α; (**M**): p-IRE1. *, *p* < 0.05; **, *p* < 0.01.

**Figure 5 cancers-15-05347-f005:**
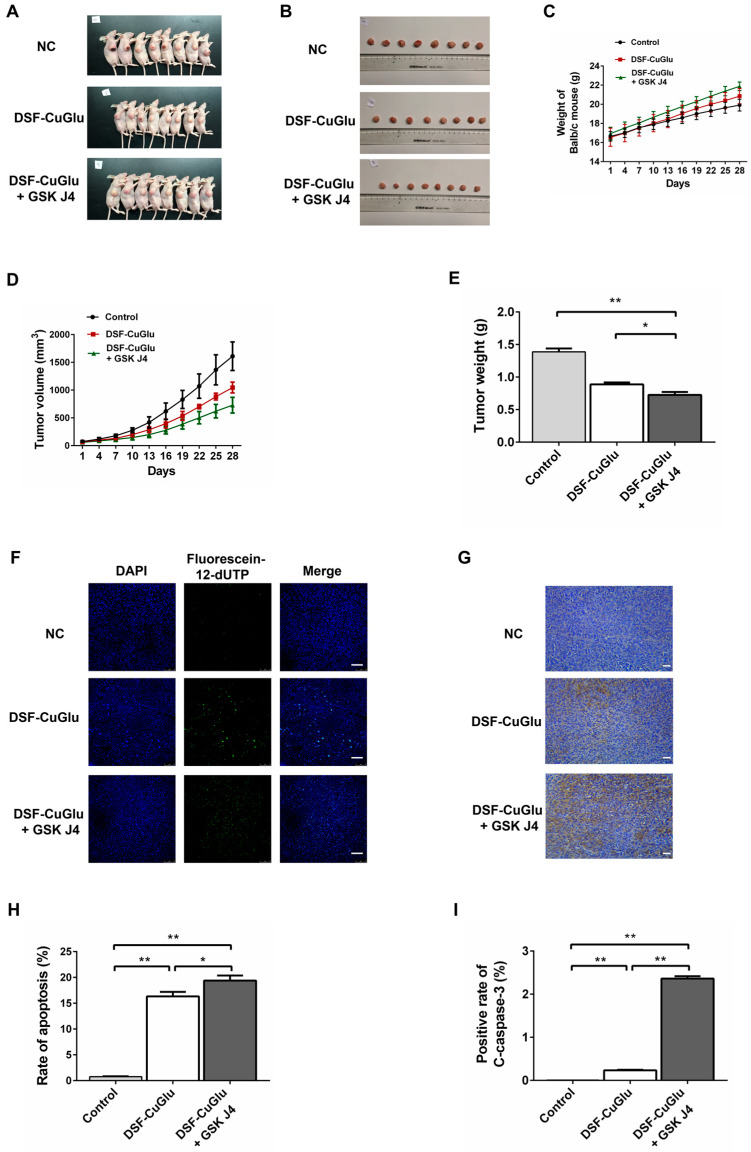
Tumor suppression effect of DSF/CuGlu and GSK J4. (**A**) Display of nude mice executed after 28 days of drug administration. (**B**) Subcutaneous neoplasms extracted from nude mice (*n* = 8). (**C**) Body weight and (**D**) tumor volume. (**E**) Comparison of subcutaneous neoplasms extracted from mice. NC, negative control; DSF, disulfiram; CuGlu, copper gluconate. (**F**) Representative images of the TUNEL assay to detect the proportion of apoptotic cells in subcutaneous neoplasms extracted from mice after drug treatment. Scale bars, 75 μm. (**G**) Representative images of immunohistochemistry to detect the expression of C-caspase-3 in subcutaneous neoplasms extracted from mice after drug treatment. Scale bars, 100 μm. (**H**) Statistical analysis of the TUNEL results in (**F**). (**I**) Statistical analysis of the immunohistochemistry results in (**G**). *, *p* < 0.05; **, *p* < 0.01.

**Figure 6 cancers-15-05347-f006:**
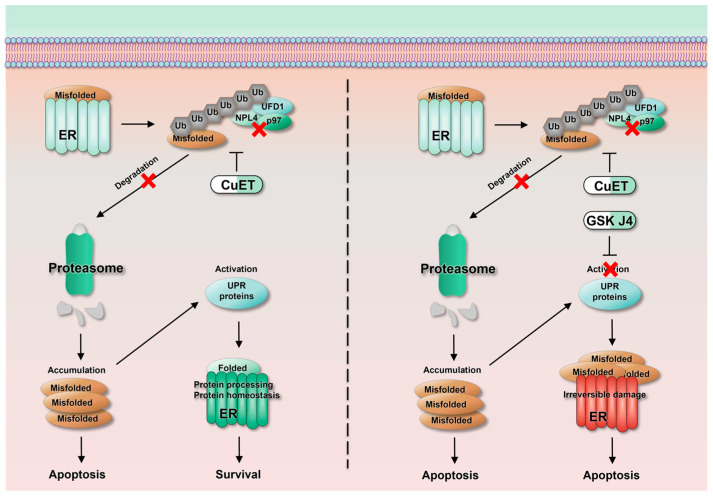
Schematic diagram of the mechanism of the synergistic effect of CuET and GSKJ4 in the treatment of ESCC. The red cross characterizes the inhibition of the signaling pathway.

**Table 1 cancers-15-05347-t001:** Correlation of mRNA expression of JMJD3 and UTX with clinicopathological characteristics of ESCC patients.

Features	Cases (*n*)	Expression of JMJD3		χ^2^	*p*-Value	Expression of UTX		χ^2^	*p*-Value
		Low	High			Low	High		
Gender									
Male	55	27 (49.1%)	28 (50.9%)	0.005	0.944	28 (50.9%)	27 (49.1%)	0.005	0.944
Female	20	10 (50.0%)	10 (50.0%)			10 (50.0%)	10 (50.0%)		
Age (years)									
<65	32	18 (56.3%)	14 (43.8%)	1.068	0.301	18 (56.3%)	14 (43.8%)	0.696	0.404
≥65	43	19 (44.2%)	24 (55.8%)			20 (46.5%)	23 (53.5%)		
Length of tumor (cm)									
≤4	37	20 (50.0%)	20 (50.0%)	0.015	0.902	22 (55.0%)	18 (45.0%)	0.644	0.422
>4	38	17 (48.6%)	18 (51.4%)			16 (45.7%)	19 (54.3%)		
Tumor location									
Upper Middle	58	28 (48.3%)	30 (51.7%)	0.114	0.735	29 (50.0%)	29 (50.0%)	0.045	0.831
Lower paragraph	17	9 (52.9%)	8 (47.1%)			9 (52.9%)	8 (47.1%)		
Differentiation									
Well + moderately	42	17 (40.5%)	25 (59.5%)	2.996	0.083	21 (50.0%)	21 (50.0%)	0.017	0.896
Poorly	33	20 (60.6%)	13 (39.4%)			17 (51.5%)	16 (48.5%)		
T Stage									
1–2	17	13 (76.5%)	4 (23.5%)	6.477	0.011 *	13 (76.5%)	4 (23.5%)	5.856	0.016 *
3–4	58	24 (41.4%)	34 (58.6%)			25 (43.1%)	33 (56.9%)		
Lymph node metastasis									
None	39	19 (48.7%)	20 (51.3%)	0.012	0.912	23 (59.0%)	16 (41.0%)	2.243	0.134
Yes	36	18 (50.0%)	18 (50.0%)			15 (41.7%)	21 (58.3%)		
TNM Staging									
Phase I–II	41	21 (51.2%)	20 (48.8%)	0.129	0.72	25 (61.0%)	16 (39.0%)	3.845	0.05
Phase III–IV	34	16 (47.1%)	18 (52.9%)			13 (38.2%)	21 (61.8%)		

*, *p* < 0.05.

**Table 2 cancers-15-05347-t002:** Synergistic effects of CuET and GSK J4 in TE10 and KYSE410 cells.

CuET + GSK J4	Combination Index at			Dose—Reduction Index at		
	IC50	IC75	IC90	IC50	IC75	IC90
TE10	0.21	0.08	0.03	7.78 ^a^	17.96 ^a^	41.45 ^a^
				12.94 ^b^	48.78 ^b^	183.89 ^b^
KYSE410	0.67	0.33	0.17	2.49 ^a^	4.49 ^a^	8.1 ^a^
				3.69 ^b^	9.10 ^b^	22.49 ^b^

^a^ fold reduction compared to single dose CuET. ^b^ fold reduction compared to single dose GSK J4.

## Data Availability

The drug screening and mass spectrometry data and the clinicopathological data of patients with ESCC generated in this study are available within the article and its Appendix A.

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
