# Peer review of "Targeting Esophageal Squamous Cell Carcinoma by Combining Copper Ionophore Disulfiram and JMJD3/UTX Inhibitor GSK J4"

_cancers, 2023, doi:10.3390/cancers15225347_

Round 1
Reviewer 1 Report
Comments and Suggestions for Authors
The manuscript presented by Yang and colleagues analyzes the effect of the combination of disulfiram (DSF) with the JMJD3/UTX inhibitor GSK J4 in various esophageal squamous cell carcinoma (ESCC) cell lines and in murine models. Through various experimental approaches, including Western blot, RT-qPCR, proteomics as well as in vivo animal models, the authors demonstrate that the combination of these two drugs has a synergistic effect in significantly reducing the aggressiveness of ECC cells and tumor growth in vivo.
Overall, the experimental design is appropriate for demonstrating the authors' hypotheses, and I really appreciate the use of in vitro, in vivo and the patients’ correlation studies. However, the manuscript requires some revisions before proceeding with publication.
Major concerns:
1. The assessment of the potential synergistic effect of the two molecules, as well as the identification of the IC50 of the molecules, was conducted on the ESCC cells following 48 hours of incubation with the drugs. In contrast, the evaluation of the potential induction of apoptosis was analyzed upon 12 hours of cell incubation (TE-10 and KYSE410) with the drugs by the Annexin V/PI assay (page 10, line 349). How do the authors correlate the effects of apoptosis observed after only 12 hours of cell exposure to the drugs with their combination following 48 hours of drug exposure? Does the drug's effect manifest within the first 12 hours of exposure and not continue, or could the percentage of apoptotic/necrotic cells significantly increase with longer incubation times? Indeed, the IC50 of CuET evaluated after 48 hours is approximately 0.6 μM (Figure 1A), while in this experiment, following a 12-hour incubation with the drug, they show a maximum of 13% of cell death (apoptosis plus necrosis).
2. Figure 3C/D. The same issue raised in point 1 also applies to the evaluation of the cell cycle. Furthermore, it would be useful to analyze the cell cycle in the ESCC cells subjected to combined treatment with the drugs to verify the hypothesis that suggested by the authors on the possible mechanism behind the observed synergistic effect of the two drugs on the cells.
3. The same issue is raised for the proteomic analyses.
4. In the graphical abstract, the authors depict the ubiquitination process of proteins and the proteasome, but neither of these two elements is ever discussed in the text. It would be appropriate to introduce a description of the role of the protein ubiquitination process and the proteasome in the context of ESCC.
Minor concerns:
1. It would be interesting if the authors could also include a summary table of the 2149 screened molecules.
2. Please increase the font size in the graphs to make the figures more easily readable.
3. Figure 2A. Why did the authors present the Real Time-qPCR data as delta Ct? In the materials and methods, they stated that they used the normalization method that involves the application of the 2-ΔΔCT formula. Therefore, the y-axis label in Figure 2A and B is not correct.
4. Figure 2C. The scale on the Y-axis is quite unusual; it interrupts at the value of 1 and then resumes with the value of 1. It also interrupts at the value of 5 and then resumes with the value of 5. Please adjust the axis labels to specify what they refer to (mRNA).
5. Figure 3E. If possible, the authors should select a representative image of the Western blot for KYSE410 cells that more accurately reflects the densitometric analysis. From the image presented, considering the loading control (GAPDH), it appears that there is a significant reduction in the levels of Caspase 3.
Reviewer 2 Report
Comments and Suggestions for Authors
The authors conducted a study to investigate the potential therapeutic efficacy of disulfiram, an alcohol-averse drug, in treating esophageal squamous cell carcinoma (ESCC). They explored the anti-tumor effects of disulfiram in combination with copper chelation (CuET) and screened a library of bioactive compounds in ESCC cells to identify potential drug combinations. Quantitative mass spectrometry was employed to analyze signaling pathway changes following drug treatment. The study revealed a strong synergistic effect between CuET and GSK J4, a dual inhibitor of JMJD3 and UTX proteins. High expression of JMJD3 and UTX was associated with advanced disease stages and poor survival outcomes in ESCC patients. In vitro and in vivo experiments demonstrated that CuET combined with GSK J4 inhibited tumor growth and induced apoptosis in ESCC cells. Furthermore, CuET triggered endoplasmic reticulum stress and unfolded protein response (UPR), which was counteracted by the addition of GSK J4. Overall, the findings suggest that combining CuET with JMJD3/UTX inhibition may offer a safe and effective treatment approach for ESCC.
I have carefully reviewed the manuscript and find it to be a valuable contribution to the field of esophageal squamous cell carcinoma (ESCC) research. However, there are several important points that need to be addressed before the paper can be considered further. Please make the following revisions and clarifications:
- Abstract Revision: Please revise the abstract to provide a more concise and clear summary of the study's objectives, methods, and key findings. Focus on highlighting the primary aim and the significance of your work in a structured manner.
- Page 2 – Line 85: The sentence, "Here, we clearly showed that combining CuET and JMJD3/UTX inhibition may be an effective drug combination for clinical use in the future," should be rephrased into a clear and concise aim sentence. For example, you can start with, "The aim of our study is to..."
- Patient Selection and Characteristics: Address the concerns regarding patient selection and characteristics. Explain the specific criteria and guidelines followed in selecting patients, especially considering the significant proportion of patients with T3-4 cancer and Stage 3-4 disease. Clarify why preoperative chemoradiotherapy was not administered and why the diagnosis of ESCC was obtained after postoperative pathology rather than through endoscopy. Additionally, describe why no patients received anti-tumor treatment and provide a sub-paragraph that outlines the clinical information of the patients.
- Statistical Power: Provide a clear discussion of the statistical power of your study. Include information on the sample population, clinical data, perioperative treatments, surgical operations, and follow-up in the Results section to ensure transparency in your analysis and findings.
The quality of English in the manuscript is generally good; however, I noticed a minor spelling error that needs to be corrected.
Round 2
Reviewer 1 Report
Comments and Suggestions for Authors
I would thank the authors to have addressed all the issue raised. However, in my opinion, they should add a sentence, in the "Materials and Methods" or in the "Results" section, to explain on which scientific bases they choose the 1 uM concentration for the 12hours treatment of their cellular models.
Reviewer 2 Report
Comments and Suggestions for Authors
The Authors have satisfactorily addressed all the points raised in the previous round of revision.
